# Synthesis and Characterization of Al Chip-Based Syntactic Foam Containing Glass Hollow Spheres Fabricated by a Semi-Solid Process

**DOI:** 10.3390/ma16062304

**Published:** 2023-03-13

**Authors:** Yong-Guk Son, Yong-Ho Park

**Affiliations:** Department of Materials Science and Engineering, Pusan National University, 2 Busandaehak-ro 63 beon-gil, Busan 46241, Republic of Korea

**Keywords:** high-strength aluminum alloy, semi-solid, glass hollow sphere, uniaxial-pressure sintering, energy absorption efficiency

## Abstract

In this study, aluminum (Al) chip matrix-based synthetic foams were fabricated by hot pressing at a semi-solid (SS) temperature. The densities of the foams ranged from 2.3 to 2.63 g/cm^3^, confirming that the density decreased with increasing glass hollow sphere (GHS) content. These values were approximately 16% lower than the densities of Al chip alloys without GHS. The Al chip syntactic foam microstructure fabricated by the semi-solid process comprised GHS uniformly distributed around the Al chip matrix and a spherical microstructure surrounded by the Mg_2_Si phase in the interior. The resulting spherical microstructure contributed significantly to the improvement of mechanical properties. Mechanical characterization confirmed that the Al chip syntactic foam exhibited a compressive strength of approximately 225–288 MPa and an energy absorption capacity of 46–47 MJ/M^3^. These results indicate higher compressive properties than typical Al syntactic foam. The Al chip microstructure, consisting of the Mg_2_Si phase and GHS, acted as a load-bearing element during compression, significantly contributing to the compressive properties of the foam. An analysis was performed using an energy-dispersive spectrometer to validate the interfacial reaction between the GHS and the matrix. The results showed that MgAl_2_O_4_ was uniformly coated around GHS, which contributed not only to the strength of the matrix, but also to the mechanical properties via the appropriate interfacial reactive coating.

## 1. Introduction

Al-based alloys are known for their high specific strength and industrial applicability [1,2,3]. However, Al alloys are difficult to use as structural materials because of their low compressive strength, energy-absorption efficiency, and wear resistance. Recently, alloys with high energy-absorption efficiency (EAE), specific strength, and low density are being used as structural materials in engineering industries through the developing Al alloys containing GHSs (AlG) [4,5]. In particular, for structural materials used in the automotive industry, alloys with energy-absorbing properties and deformation resistance are needed for passenger/goods safety during collisions. AlG foam is an alloy that possesses high energy absorption properties through GHS and has been widely used as a structural material in recent years [6,7].

AlG foam is an alloy in which GHSs are uniformly distributed within the metal matrix. These alloys are widely known for their low density and high EAE [8,9]. Materials with a high EAE are particularly advantageous under compressive loading as the GHS becomes a load-bearing element in compression. In addition, AlG foam exhibits a large stress plateau in the compression state, which leads to densification and a high EAE in the Al alloy [10,11].

AlG foam exhibits better mechanical properties than existing alloys because its high-energy absorption properties can delay deformation. The high energy-absorption properties relate to the microstructural properties of AlG foam, which can be defined by various properties such as matrix alloy strength, chemical composition, GHS strength, matrix/GHS volume ratio, interfacial diffusion, and relative density. Recently, many researchers analyzed the compressive properties of AlG foam, which helped in analyzing the microstructural features that affect the mechanical strength of alloys and, thus, explain their mechanical behavior.

For instance, Katona et al. used dynamic mechanical analysis, finite element methods, and elasticity-based analytical calculations to analyze the compressive properties of Al and Al-Si syntactic foams [12]. Their model is based on a matrix of fracture mechanisms. Their model analyzed the fracture mechanism according to the matrix and demonstrated that the compressive strength and energy absorption can be improved. Szlancsik et al. generalized the compression curves of Al alloys containing pure Fe metallic hollow spheres through compression tests and analyzed properties such as structural stiffness, yield strength, plateau strength, and energy absorption [13]. They revealed that their alloys were affected by the matrix microstructure and relative density. Jung et al. fabricated Al-Mg matrix-based syntactic foams using a stir casting process and identified the associated changes in X-ray diffraction, EPMA, and mechanical properties. In particular, they showed that the compressive properties and energy absorption may be increased due to the interfacial reaction between glass hollow sphere and matrix [14]. Wang et al. [15] studied the compressive behavior and energy absorption properties of Al foam with different cell sizes. Several studies proposed that Al foam improves the EAE but has a lower compressive strength than conventional alloys. In order to improve the compressive strength of such Al foams, research was actively conducted by changing the manufacturing process of AlG foams and mixing bimodal GHS.

For instance, Kemeny et al. fabricated high-performance bimodal composite metal foams via a low-pressure permeation process to increase compressive strength. The produced metal foams were analyzed by compression testing and showed a strength of about 30~60 MPa. They proved that the mechanical properties of the metal foam improved with GHS filling ratio [16]. Su et al. fabricated Al matrix bimodal syntactic foam by mixing 1.0~4.0 mm alumina GHS using a stir casting method. The produced metal foams exhibit compressive strengths of approximately 60~80 MPa. It was found that when GHS of different size ranges are mixed in the same volume, the mechanical properties of the bimodal syntactic foam are highly dependent on the average diameter of the GHS and that the mechanical properties decrease as the GHS content increases [17]. Unlike common manufacturing methods, Bolat et al. produced Al matrix syntactic foam using a fully automated low-temperature chamber die casting process. The produced Al syntactic foam exhibits compressive strengths of approximately 20–80 MPa. They identified the effects of heat treatment on the compressive properties and fracture behavior of the samples, demonstrating that heat treatment alters the fracture mode of the samples [18]. However, the compressive strength of currently manufactured Al syntactic foam is below about 100 MPa and still exhibits low mechanical properties.

This study focuses on the fabrication of alloys that can simultaneously improve the strength and EAE of Al chip alloys with high compressive properties through previous studies [19]. Several studies suggested that the spheroidized microstructure of Al alloys produced using the SS process improves compressive strength [20,21]. In addition, alloys fabricated by chips have the advantage that samples can be produced at lower temperatures than bulk alloys [22]. Based on the results of these studies, the Al alloy processed by chip was sampled by powder metallurgy, and an Al chip matrix was spheroidized via an SS process to further improve the mechanical properties of the Al chip alloys. GHSs were added to the interfacial reaction between GHS and the Al chip to improve the EAE of the Al chip alloy, and the compressive behavior was investigated.

## 2. Materials and Methods

### 2.1. Fabrication of Syntactic Foams

The AlG foam matrix was designed by melt casting an Al-8Zn-6Si-4Mg-2Cu alloy, and the fabricated Al alloy was machined into Al chips with a width and length of approximately 1 mm or less (Figure 1). Al chips were processed in a metal crusher. It was sieved with a 500 µm sieve to obtain uniformly sized AI chips. AlG foam was synthesized using a uniaxial-pressure sintering process. The fabricated samples contained 50, 60, and 70 vol.% Al chips and 5, 10, and 15 vol.% GHS. The excluded volume fractions of Al chip and GHS were mixed with Al powder (purity = 99.7%, average particle = 5 μm) to bridge the gap between the GHS and Al chip. To uniformly distribute the GHS and pure Al powder in the Al chip matrix, 3 mL of heptane was added as a binder and mixed in a 3D mixer for 24 h. In the first process, the mixed samples were held for 30 min after a temperature increase of 10 °C/min till they reached 450 °C under a pressure of 120 MPa. The pressure was removed in a secondary process. In the second process, the temperature was increased to 560 °C at a rate of 10 °C/min and maintained for 30 min. The reason for removing pressure from the secondary process was that it is difficult to spheroidize the inside of the Al chip matrix due to increasing temperature while maintaining pressure. These results were analyzed according to a previous study [22]. The fabricated foams were denoted as Al50G5, Al60G5, Al70G5, Al50G10, Al60G10, Al70G10, Al50G15, Al60G15, and Al70G15, where the numbers after Al represent the volume fractions of Al chips and the numbers after G represent the volume fraction of GHS (Table 1). A schematic of the foam-production process is shown in Figure 1b. The thermocouples were installed inside the mold to measure the temperature (Figure 1c). GHS size was measured using a particle size analyzer (Beckman Coulter Ls 13 320, Brea, CA, USA). Figure 2 shows the GHS particle size distribution. The mean GHS diameter was 16.86 μm d10, d50, and d90 diameters were 6.34, 15.8, and 29.4 μm, respectively. The d10, d50, and d90 were the maximum particle diameters that contained up to 10%, 50%, and 90% of the particles, respectively. The Al alloy, GHS, and Al powder chemical compositions were measured using optical emission spectroscopy (OES; OES-6000, Shimadzu, Kyoto, Japan), and the results are presented in Table 2 and Table 3. The density was measured as the average of five test results of each sample using the Archimedes method.

### 2.2. Microstructural Analysis

For microstructural characterization, the GHS was mounted using a hot-mount machine, and the fabricated 30-pie size AlG foam was cut from the center into a size of 10 × 10 × 10 mm. The samples were polished using SiC paper and mechanically polished using 3 and 1 μm suspension. The microstructure of the GHS was analyzed using field-emission scanning electron microscopy (FE-SEM, S-4800 Hitachi, Tokyo, Japan) equipped with an energy-dispersive X-ray spectroscopy (EDS) detector (Horiba 51-ADD0014, Horiba, Kyoto, Japan). The microstructure of the syntactic foam was analyzed using optical microscopy (OM, Leica DM2700M RL/TL, Leica, Wetzlar, Germany). For phase analysis of AlG foam, diffraction data were collected using high-resolution X-ray diffraction (XRD, Empyrean, Panalytical, Malvern, UK). Cu-Kα radiation (wavelength: 1.540598 Å) was used to obtain diffraction peaks. Operating conditions were 45 kV and 20 mA, and the diffraction range was approximately 10–90° in θ–2θ mode.

### 2.3. Mechanical Property Testing

For the compressive tests, 4 × 4 × 8 mm specimens were prepared according to ASTM E9. The compressive tests were carried out five times for each sample. The compressive properties of the foam were estimated using a universal testing machine (Shimadzu TCE-N300-CE, Kyoto, Japan) equipped with a 50 kN load cell with an applied strain of 1.0 × 10^−3^·S^−1^. The compressive strain was evaluated using strain gauges in the elastic deformation region. The hardness test was performed using a micro-Vickers hardness tester (HMV-G31ST, Shimadzu, Kyoto, Japan) with a pressing force of 0.004 kgf and a dwell time of 10 s. A diamond with a face angle of 136° and a squared-based pyramidal shape based on ASTM E92-17 was used as the indenter. The indentation depth (hV) was calculated using the following equation: [23]
hV = 0.143 × dV(1)
where dV is the mean length of the indentation diagonal in μm.

## 3. Results and Discussion

Figure 3 shows an SEM image and electron emission spectroscopy element maps of GHS. Figure 3a shows that GHS particles were spherical, and some spheres represented GHS satellite particles attached to the surface, while the particles in Figure 3b represented a single GHS. The detected elements were Si, O, Ca, Na, Mg, Al, K, Ti, and Fe. The elemental composition estimated based on EDS area analysis showed low Al, K, Ti, and Fe concentrations in GHS (Table 4). GHS primarily comprises of SiO_2_, Na_2_O, CaO, Al_2_O_3_, and MgO [24,25].

Figure 4 shows the evolutions of the relative densities of the AlG foams with increasing volumes of GHSs. The densities were measured from local parts from the bottom, middle, and top of each sample, and it was found that the density was uniform throughout for every specimen tested. The density decreased with increasing the GHS volume fraction. The theoretical density of AlG foam (D_AlG foam_), and Al chip (D_Al chip_), can be predicted using the following equation.
D_Al chip_ = (D_Al_ × C_Al_) + (D_Zn_ × C_Zn_) + (D_Si_ × C_Si_) + (D_Mg_ × C_Mg_) + (D_Cu_ × C_Cu_)(2)
D_AlG foam_ = (D_Al_ × C_Al_) + (D_Zn_ × C_Zn_) + (D_Si_ × C_Si_) + (D_Mg_ × C_Mg_) + (D_Cu_ × C_Cu_) + (D_GHS_ × C_GHS_)(3)
where D_x_ and C_x_ represent the actual density and volume fraction of element x in the composition, respectively. The theoretical densities of Al chip and AlG foam calculated by Equations (2) and (3) are shown in Figure 4 for comparison with the measured values. The theoretical densities of AlG foam mixed with 100 vol.% Al chip alloy and 5, 10, and 15% GHS were 2.78, 2.66, 2.54, and 2.43 g/cm^3^, respectively. The relative density of AlG foam was reduced by about 6–16% compared to the theoretical density of the 100% Al chip alloy. These results indicate that the relative density of the AlG foam was reduced by the relatively light GHS mixture. The relative densities of AlG foams containing 5, 10, and 15 vol.% showed 98.8%, 98.7%, and 96.2% relative to the theoretical density of the AlG foam. The relative density of the AlG foam was shown to be close to the theoretical density, and the partially reduced relative density of the 15 vol.% AlG foam can be explained as unintended microporous formation during the process [26].

The Vickers hardness test was conducted to confirm the effect of GHS content on the Al foam mixed with the Al chip alloy and GHS. Figure 5a shows the schematic of the hardness measurement area. Hardness measurements were taken in the α-Al region of the Al chips and in the Al powder and GHS interface region to determine the change in the mechanical properties of the Al matrix with the change in GHS content. The indentation depth calculated using Equation (1) was estimated to be 4–6 μm. Figure 5b shows the evolutions of the hardness of the AlG foam with increasing volume fraction of GHS increased. Hardness decreased little with volume fraction of GHS. The matrix of the Al chip alloy shows a hardness of 158.1 HV and the matrix hardness of the AlG foam by GHS content was about 154.2–156.3 HV. The Al chip alloy matrix and the AlG foam matrix did not significantly differ in hardness. This is expected because of an appropriate interfacial reaction between the GHS and the Al matrix. The appropriate interfacial reaction between the GHS and matrix affects the mechanical properties [27,28]. Based on these results, it was presumed that the interfacial bonding between the GHS and the Al matrix of AlG foam was well done and the hardness was not reduced.

In addition, the Mg and Si atoms in this alloy crystallized according to the composition ratio and Mg_2_Si phase led to precipitation strengthening [29]. Factors contributing to the alloy strength of the Al chips produced by the SS process include matrix spheroidization and precipitated Mg_2_Si phase formation. As shown in Figure 5b, the contribution of matrix spheroidization and precipitation strengthening is similar for GHS-containing and Al chip alloys, suggesting no reduction in hardness with GHS content.

Figure 6 shows the OM images of each sample. GHS is mainly distributed at the Al powder and Al chip interface. GHS dispersion was most uniform in Al50G10, Al60G10, and Al70G10. In Al50G5, Al60G5, and Al70G5 with low GHS content, GHS is distributed more uniformly as the Al chip ratio increases. A microstructure with locally distributed small amounts of GHS appeared as the Al powder ratio increased. In Al50G15, Al60G15, and Al70G15 with a higher GHS content, GHS aggregated in Al powder of the relatively lower Al powder content than in other foams. These results were explained as the volume fraction of Al powder that bridged between the GHS and Al chip. The lower the volume fraction of Al powder, the more GHS agglomerates within the area of Al powder for samples containing relatively high GHS volume fraction [30,31]. Therefore, in Al50G15, Al60G15, and Al70G15, the area filled with Al powder and GHS was minuscule, and the GHS agglomeration rate was higher.

Figure 7 shows the OM images of Al chip matrix microstructures in Al50G5, Al60G5, Al70G5, Al50G10, Al60G10, Al70G10, Al50G15, Al60G15, and Al70G15 foams. The microstructure of the Al chip matrix was spheroidized according to the SS process. Previous studies confirmed that the Al_5_Cu_2_Mg_8_Si_6_ phase became the liquid phase via DSC at about 450–470 °C [19]. Small amounts of liquid are essential in the spheroidization of the Al chip matrix. Under 120 MPa pressure, a small volume of liquid was squeezed onto the surface of the dendrite. The Al matrix was then spheroidized by rheology and the Ostwald ripening phenomenon after a longer holding time at 560 °C [32]. The spheroidized Al chip matrix suppresses dislocation pile-up and dislocation progression [33,34,35]. This strengthening effects contribute to the alloy strength of the AlG foam.

Figure 8a shows the XRD patterns of Al chips with noticeable α-Al, Mg_2_Si, Al_2_Cu, and Si peaks. These results were consistent with the phases formed during the matrix alloy fabrication process in previous studies [36]. Figure 8b–d shows the XRD patterns of the AlG foam with peaks of the MgAl_2_O_4_ phase and the α-Al, Mg_2_Si, Al_2_Cu, and Si phases. The MgAl_2_O_4_ phase was presumed to be formed by the interfacial reaction between the matrix alloy and GHS.

Figure 9a shows the EDS point profiling of the GHS before sintering, and Figure 9b shows the EDS point profile analyzed in the shell region of the GHS after sintering. The results of the EDS point profiling analysis are shown in Table 5 and Table 6. The EDS point profiles in Figure 9a and Table 5 showed a 28.05 wt.% distribution of Si elements in the GHS. On the other hand, the EDS point profiles in Figure 9b and Table 6 showed sparse distribution of 1.19 wt.% of Si elements in the GHS shell. The major elements constituting the GHS were Si, O, Ca, Na, Mg, and Al, as shown in Table 5. After sintering, the concentrations of Si, O, Ca, and Na decreased, while those of Al and Mg increased. These results suggest that Mg and Al elements were diffused into the GHS shell during sintering in Al alloy chips. In addition, Si and most trace elements were estimated to have been reduced at high temperatures.

Figure 10 shows the EDS mapping and line profiles used to analyze the qualitative composition of the GHS that reacted with Al60G10. As shown in Figure 10b,c, the Al-rich regions appeared uniform in the matrix, and Si rich regions surround the GHS shell. This indicated that Si in the GHS diffused outside and concentrated at the boundary between the matrix and GHS. Figure 10d shows the detailed enrichment of Mg in the shell region of the GHS. In addition, as shown in Table 5 and Table 6, Mg was detected at higher concentrations at the GHS matrix interface than inside the GHS. The oxygen (O)-rich regions overlapped with segregated regions of the Mg and Si solutes. The oxygen around the GHS was presumed to be present owing to the oxidation of Mg. It is known that oxidation by Mg can easily occur at high temperatures in Mg-containing Al alloys [37,38]. No other trace elements present in the GHS were detected, as they were expected to have been depleted by solute diffusion. The Mg- and Si-rich regions were associated with the Mg_2_Si phase around the GHS shell, which appear to have been formed by interfacial diffusion. At low temperatures, Mg can dissolve in Al [39]. Thus, Mg can exist as a solid solution in α-Al or as an intermetallic compound that can react with other elements. These results confirmed the formation of the Mg_2_Si phase in the GHS shell region by the EDS point profile analysis of the red square region in Figure 10f. The formation of the Mg_2_Si phase can also be validated by a line profiling analysis at the GHS- matrix interface. In Figure 10g, a high Al content was seen beyond the GHS boundary, while high concentrations of O and Mg were uniformly distributed in the interface region and decreased past the GHS shell. In contrast to Mg, the concentration of Si in the matrix increased. These results indicate that the Si concentration gradually increased toward the matrix by interfacial diffusion. Once the Si concentration reached a certain level in the matrix, the Si phase precipitated. Mg was concentrated near the interface region, and high Mg concentrations reacted with the precipitated Si to form a Mg_2_Si phase in the GHS shell. The Gibbs free energy associated with the reaction between the GHS and matrix was the driving force of the interfacial reaction in the GHS shell region. The most likely interfacial reactions were calculated by referring to the thermodynamic database as follows: [40]
(4)$$2{\mathrm{Al}}_{\left(l\right)}+{\;\mathrm{Mg}}_{\left(l\right)}+2{\mathrm{SiO}}_{2\left(s\right)}\to {\mathrm{MgAl}}_{2}O_{4\left(s\right)}+2{\mathrm{Si}}_{\left(s\right)},\;\Delta G_{470{}^{\circ}C}=-448.1\mathrm{kJ}$$
(5)$$2{\mathrm{Mg}}_{\left(l\right)}+{\;\mathrm{Si}}_{\left(s\right)}\to {\mathrm{Mg}}_{2}{\mathrm{Si}}_{\left(s\right)},\;\Delta G_{470{}^{\circ}C}=-61\mathrm{KJ}$$

It was speculated that SiO_2_, a major component of the GHS, was highly reactive with Mg and Al at the temperature needed to produce AlG foam. Through Equations (4) and (5) for interfacial reactions calculated with reference to the thermodynamic database, the interfacial reactions form spinel MgAl_2_O_4_ in the GHS shells and Mg_2_Si in the regions surrounding the shells. As SiO_2_ reacted with Al and Mg, the Si in the GHS may have diffused. As shown in Figure 10g, Si precipitated from the matrix region around the GHS shell. These results suggest that Si diffused through the MgAl_2_O_4_ grain boundary and precipitated in the matrix next to the GHS shell [41]. Furthermore, as the GHS content of the AlG foam increased, the Mg content did not change. In Equation (4), Mg reacted continuously with Si around the GHS shell to form Mg_2_Si when MgAl_2_O_4_ no longer formed.

The quasi-static compressive stress–strain curves of the AlG foam according to the GHS and Al chip content are shown in Figure 11a–c. The AlG foam curve initially exhibited linear elastic and plateau plastic regions, in which the stress monotonically increased with increasing strain and, finally, rapidly decreased. The maximum stress in the AlG foam occurred at the start of the plastic deformation, as shown in Figure 11a–c. The yield strength of AlG foams containing the same volume fraction of GHS increased as the Al chip content increased. The compressive strengths of Al70G5, Al60G5, and Al50G5 were 288.7, 264.5, and 230.5 MPa, respectively, while those of Al70G10, Al60G10, and Al50G10 were 281.5, 251.3, and 225.4 MPa, respectively. The compression curves of the AlG foam with 5 and 10 vol.% GHS showed an increase in plateau stress and a delay in the onset of densification with increasing GHS content, as shown in Figure 11a,b. In contrast, the compression curve of the AlG foam with 15 vol.% GHS showed that the plateau stress decreased with increasing GHS content and densification proceeded faster, as shown in Figure 11c. Al50G15, Al60G15, and Al70G15 foams had a high GHS content, and GHS aggregation occurred regardless of the Al powder, which played a bridging role. GHS agglomeration was expected to cause a heterogeneous phenomenon in the loading transfer of the AlG foam—resulting in reduced plateau stress and accelerated densification despite the increased GHS content. The compressive properties of all the specimens are listed in Table 7. The energy absorption ($W=\int_{0}^{\varepsilon_{d}}\sigma d\varepsilon$), one of the key features of the compressive properties, was measured using the tangent line technique, presented by Basit and Cheon, up to the densification strain of the AlG foam [42]. Another key feature was the energy-absorption efficiency calculated using Equation (6) for a given deformation [43].
(6)$$W_{e}=\frac{W}{\sigma max^{\varepsilon }}\times 100\;$$
where $W_{e}$ is the EAE, W is the energy absorption, and $\sigma max^{\varepsilon }$ is the maximum compressive stress within the strain range. The energy absorption and EAE of the AlG foam increased as the GHS content increased because both the plateau stress and densification strain increased as the GHS content increased.

The improved compressive properties of the AlG foam were, presumably, due to the interfacial reactions between the GHS and the matrix and other strengthening effects. Owing to the specific interfacial reactions, the increase in compressive properties is explained by two factors. First, the MgAl_2_O_4_ formed by the interfacial reaction improved the interfacial bonding between the matrix and GHS. Second, the fracture strength of MgAl_2_O_4_ coated on the GHS surface may have been higher than that of the GHS shell. Appropriate interfacial reactions strengthened the interfacial bonding of the AlG foams, which suggests that the strength of the AlG alloy was enhanced by MgAl_2_O_4_, which uniformly formed on the GHS surface.

The Mg_2_Si phase had a low density and excellent mechanical properties and was a strengthening phase mainly applied to Al alloys. The Mg_2_Si phase precipitated in the AlG foam matrix and around the GHS shell contributed to the increase in the strength of the AlG foam alloy through a secondary-phase strengthening mechanism.

The alloy strength of the AlG foam used in this study exhibited three strengthening effects. First, Mg in the Al chips can improve the compressive properties by uniformly coating the GHS shell with an interface reaction layer, spinel MgAl_2_O_4_, through an interface reaction. Second, Mg reacts with Si to precipitate the Mg_2_Si phase around the matrix and GHS shell potentially improving the compressive properties as a secondary strengthening phase. Finally, the mechanical properties can be improved by dislocation pile-up via spheroidizing inside the Al chip. The interconnection of these three reinforcing effects was presumed to have simultaneously improved the compressive properties and energy absorption of the AlG foam. As a result, AlG foams had lower compressive strength but higher EAE than Al chip alloys.

Figure 12 shows the predicted and measured compressive strengths of AlG foam at different GHS contents. The predicted compressive strength of AlG foam can be estimated by the rule of mixtures [44]:(7)$$\sigma_{AlG\;foam\;com}=V_{GHS}\sigma_{GHS\;com}+V_{Al\;chip}\sigma_{Al\;chip\;com}+V_{Al\;powder}\sigma_{Al\;powder\;com}$$
where $\sigma_{AlG\;foam\;com}$ is the compressive strength of the AlG foam, V is the respective volume fractions of the mixed material, and σ is the respective compressive strength of the mixed material. The predicted compressive strength of AlG foam decreased with increasing GHS content. This behavior was similar to the measured compressive strength. The predicted values for AlG foam containing 5 and 10 vol.% GHS appear to agree with the measured values. These results indicate that there were few defects inside the AlG foam, and the three strengthening effects (Spheroidization of Al chip microstructure, Mg_2_Si reinforcing phase, and interfacial bonding by the MgAl_2_O_4_ spinel phase) of AlG foam described above were appropriately well applied. However, AlG foam containing 15 vol.% GHS appears to have had a lower measured compressive strength than the predicted one. These results were expected because of local micro pore formation and GHS aggregation inside the AlG foam.

Figure 13 shows the GHS fracture behavior of the Al60G10 foam according to the strain. As shown in Figure 13a,b, the GHS maintained its spherical shape without fracture or deformation up to 4% strain. Figure 13c shows that cracking began at 8% strain and the crack propagation moved from the GHS to the matrix. As shown in Figure 13d, cracks propagated over most of the GHS surface at 12% strain and in the matrix. Finally, in Figure 13e, both the matrix and GHS were destroyed because of specimen densification. These results can be explained by one of the strengthening effects of the AlG foam’s increased interfacial bonding due to the MgAl_2_O_4_ coating on the GHS surface. Weak interfacial bonding between the GHS and metal matrix in tensile and compression tests caused GHS to separate from the matrix, creating empty cavities [45,46]. When the GHS and metal matrix were separated, the uniform load transfer phenomenon was eliminated, and stress was concentrated locally in the matrix, thereby significantly impacting the mechanical properties. As shown in Figure 13, the interfacial bonding was high because of the MgAl_2_O_4_ formed by the interfacial reaction between the GHS and the matrix, and there was no separation between the GHS and matrix. These results were presumed to be one of the leading causes for the improved energy absorption of the AlG foam.

Figure 14 shows crack propagation with GHS content. The *y*-axis indicates the direction of compressive loading on the microstructure. As shown in Figure 14, crack propagation was observed along the direction of the yellow arrows. The white arrows in Figure 14b indicate the presence of the GHS. Cracks propagated along the GHS and Al metal matrix in the direction perpendicular to the compressive load. As shown in Figure 14a, the Al60G5 foam distributed the load at the interface between the GHS and the Al metal matrix, causing plastic deformation of the Al matrix. After reaching yield deformation, micro-cracks occurred at the interface between the Al metal matrix and GHS, and the cracks propagated to adjacent interfaces. On the other hand, as shown in Figure 14b, the Al60G10 foam showed a higher GHS content, resulting in multidirectional crack propagation along the GHS, potentially producing uniform load transfer inside the foam and slowing the growth of major cracks inside the metal matrix. These results indicate that at low GHS content, crack propagation occurred in the Al matrix adjacent to the GHS, with high compressive strength but low EAE. On the other hand, if the GHS content was high, the growth of cracks propagating along the GHS may have resulted in high EAE, but the compressive strength was expected to decrease.

Figure 15 shows the energy absorption versus compressive strength of the Al syntactic foam in this and other studies [6,14,40,43,47,48,49,50,51]. The AlG foam with 10 vol.% synthesized in this study had the highest compressive strength and the third highest energy absorption rate among those reported in the literature. However, AlG foams containing 5 and 10 vol.% had high compressive strengths but normal energy absorptions. These results mean that it is possible to synthesize materials with high compressive strengths and energy absorptions of the Al foam. It is also expected that a uniaxial-pressure sintering system can achieve the production of functional materials whose stability is ensured.

## 4. Conclusions

1. The semi-solid process using uniaxial-pressure sintering successfully produced Al chip-based syntactic foam with a volume fraction of up to 15% of GHS. The microstructure of spheroidized Al chips produced by the semi-solid process was characterized by higher strength than that of ordinary Al. For uniaxial pressure sintering, the temperature was raised from 0 to 450 °C and 120 MPa was applied during sintering at 450 °C for 30 min. Then, after removing the pressure from the process, an Al chip-based syntactic foam could be produced with a sintering temperature of 560 °C and a sintering time of 30 min.

2. The average density of the produced foam was up to 16% lower than that of the Al chip alloy without GHS. It also had a relative density that was close to a maximum of 98% relative to the theoretical density of the synthesized foam. GHS were distributed between the Al chip and the chip in the microstructure, and uniform spheroidization of the Al chip matrix was observed. The spheroidization of the matrix appeared to be caused by a small amount of liquid being squeezed onto the dendrite surface, followed by an Ostwald ripening mechanism.

3. The interfacial reactions between the GHS and the Al chip alloy were investigated (by phase transformation) by thermodynamic simulations and microstructural characterization. Aluminum and magnesium were formed in the GHS shell by the phase transformation to MgAl_2_O_4_ through the reaction with SiO_2_ in the GHS with a composite chemical composition. The spinel MgAl_2_O_4_ formed during the interfacial reaction between the GHS and the matrix can potentially strengthen the interfacial bond. Furthermore, while MgAl_2_O_4_ was being formed, the Si solute was released, by the reduction in GHS and it reacted with Mg in the matrix to form a Mg_2_Si phase around the GHS shell. The precipitation of Mg_2_Si improved the mechanical properties of the foam.

4. Foam containing 10 vol.% volume fraction of GHS in compressive properties exhibits compressive strengths of up to 288 MPa and energy absorption of 47 MJ/M^3^. These results showed improved compressive properties and energy absorption compared to conventional Al syntactic foams due to the three reinforcing effects of the spheroidized Al chip matrix, spinel MgAl_2_O_4_ by interfacial reaction, and Mg_2_Si as the secondary phase. On the other hand, foam containing a 15 vol.% volume fraction of GHS showed low compressive properties because the agglomerates of the GHS did not result in uniform load transfer.

## Figures and Tables

**Figure 1 materials-16-02304-f001:**
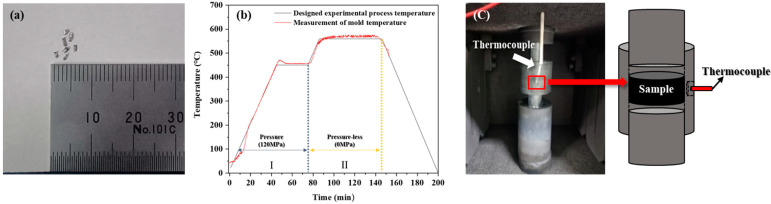
(**a**) Al chip size measurement; (**b**) Schematic of AlG foam process temperature and actual mold temperature measurements; (**c**) Mold temperature measurement method.

**Figure 2 materials-16-02304-f002:**
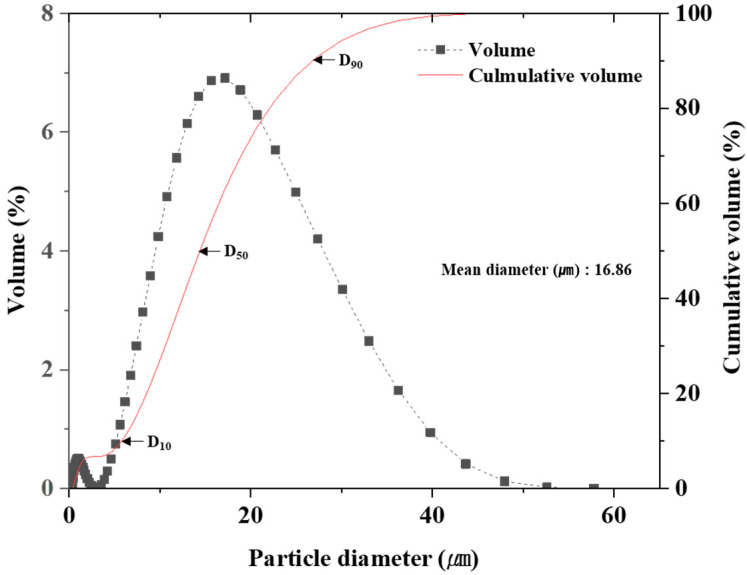
Particle size distribution and volume percentile of GHS.

**Figure 3 materials-16-02304-f003:**
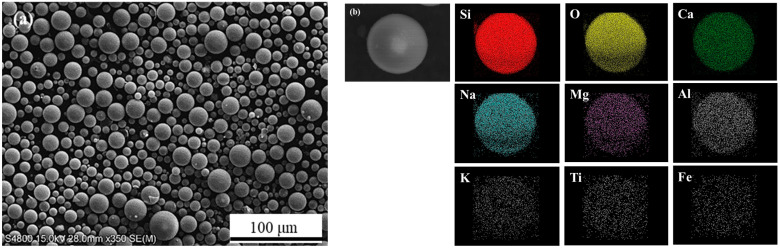
(**a**) SEM image of GHS; (**b**) EDS map for the microstructure of the GHS.

**Figure 4 materials-16-02304-f004:**
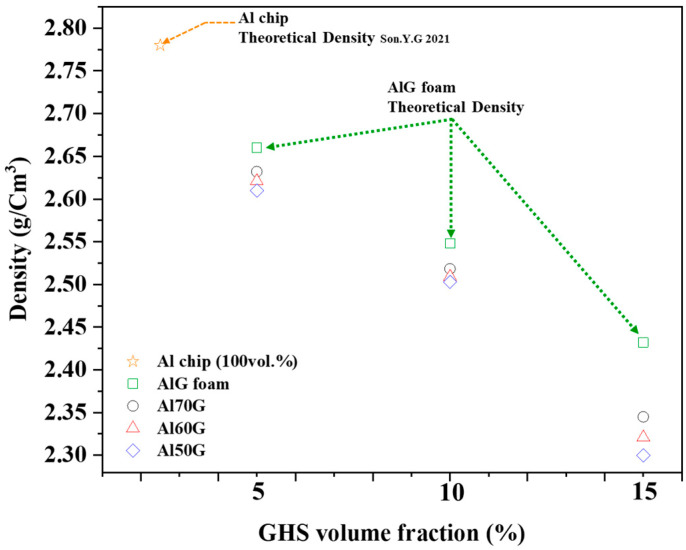
Variation of theoretical and relative densities of samples by volume fraction of GHS [19].

**Figure 5 materials-16-02304-f005:**
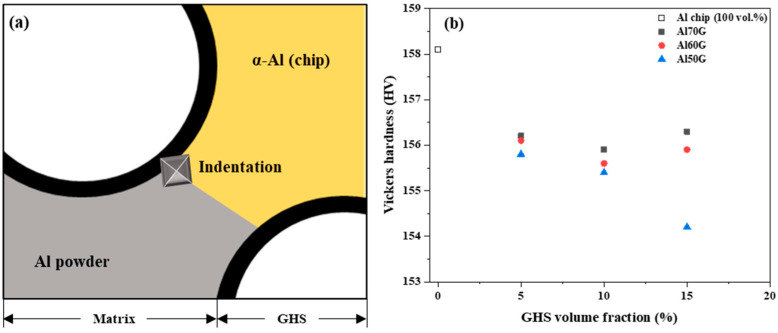
(**a**) Region of hardness measurement; (**b**) hardness variation in Al chip and Al powder matrix with volume fraction of GHS.

**Figure 6 materials-16-02304-f006:**
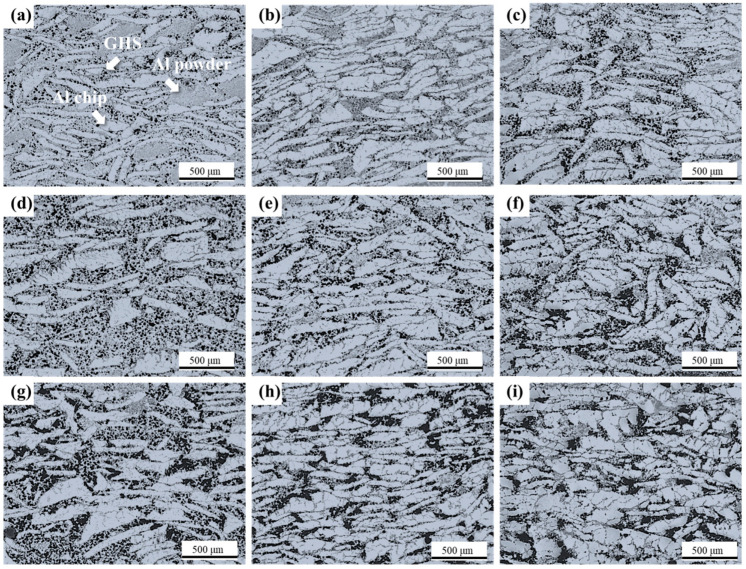
50x magnification OM image of microstructures of each sample; (**a**) Al50G5, (**b**) Al60G5, (**c**) Al70G5, (**d**) Al50G10, (**e**) Al60G10, (**f**) Al70G10, (**g**) Al50G15, (**h**) Al60G15, (**i**) Al70G15.

**Figure 7 materials-16-02304-f007:**
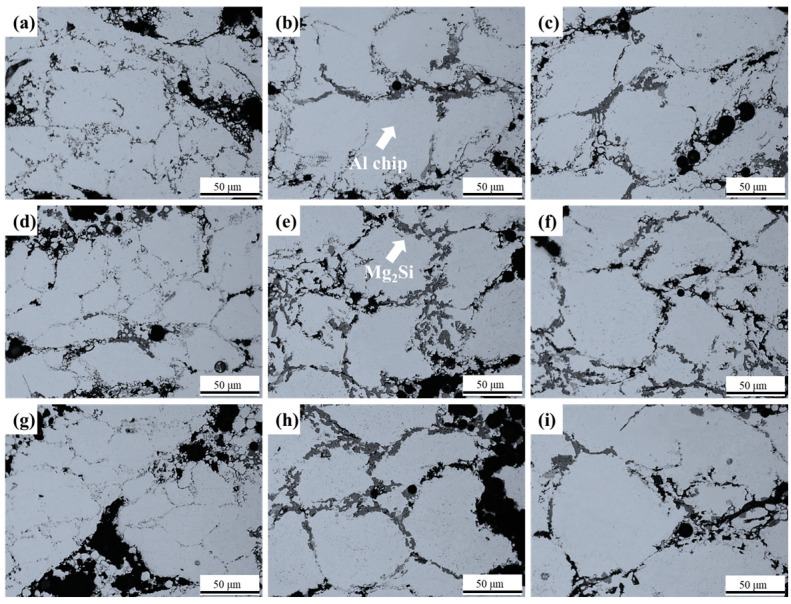
500x magnification OM images of microstructures of each sample; (**a**) Al50G5, (**b**) Al60G5, (**c**) Al70G5, (**d**) Al50G10, (**e**) Al60G10, (**f**) Al70G10, (**g**) Al50G15, (**h**) Al60G15, (**i**) Al70G15.

**Figure 8 materials-16-02304-f008:**
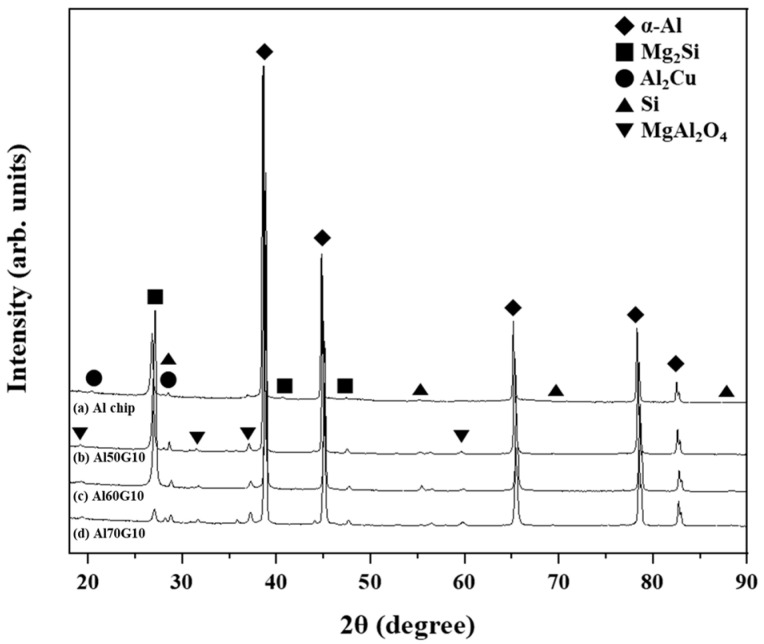
XRD patterns of Al chip alloy and AlG foam: (a) Al chip; (b) Al50G10; (c) Al60G10; (d) Al70G10.

**Figure 9 materials-16-02304-f009:**
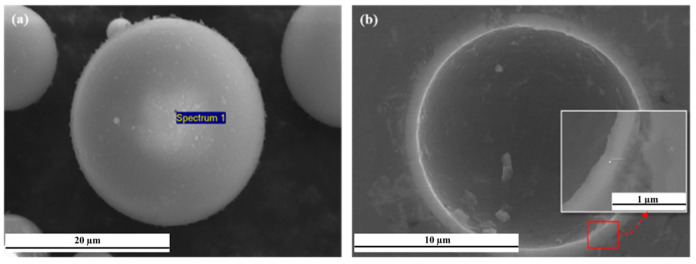
SEM and corresponding EDS point profiling of the GHS: (**a**) SEM of the GHS; (**b**) SEM images of the GHS after sintering.

**Figure 10 materials-16-02304-f010:**
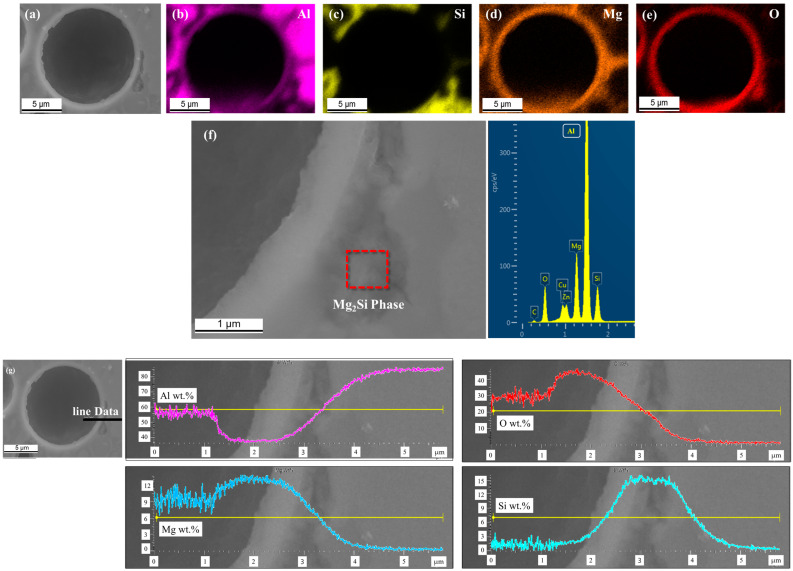
SEM and corresponding EDS results for Al60G10 foam: (**a**) SEM image of GHS; (**b**) Al-k map; (**c**) Si-K map; (**d**) Mg-k map; (**e**) O-k map; (**f**) EDS point profiling around GHS shell; (**g**) EDS line profile image of GHS-matrix interface.

**Figure 11 materials-16-02304-f011:**
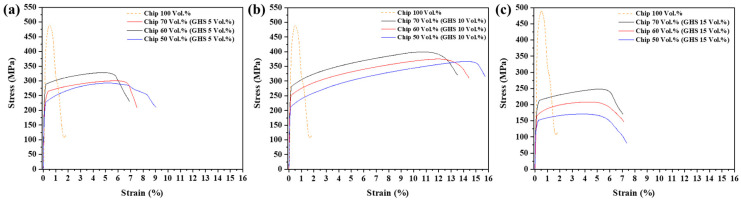
Compressive stress-deformation curves for Al chip alloys and AlG foams: (**a**) Al(50, 60, and 70)G5; (**b**) Al(50, 60, and 70)G10; (**c**) Al(50, 60, and 70)G15.

**Figure 12 materials-16-02304-f012:**
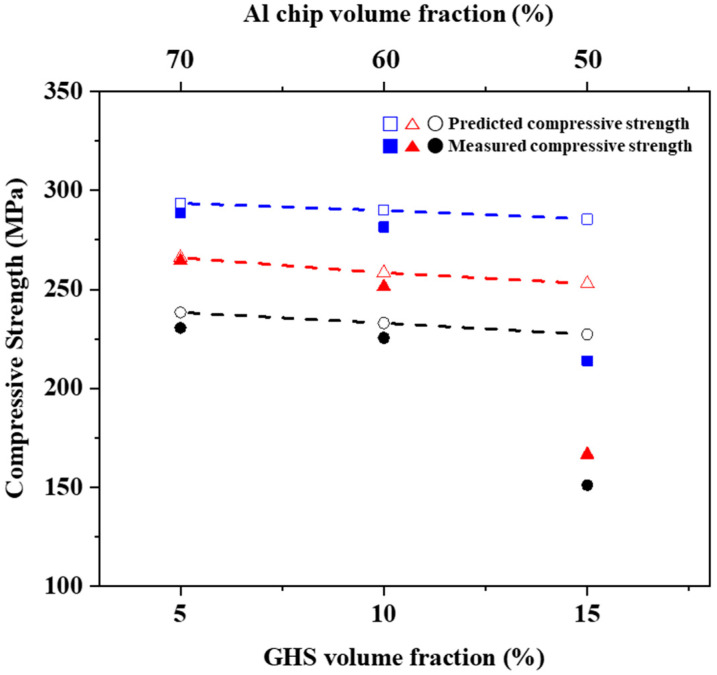
Predicted (dotted lines) and measured (dots) compressive strengths of AlG foam with different GHS contents.

**Figure 13 materials-16-02304-f013:**
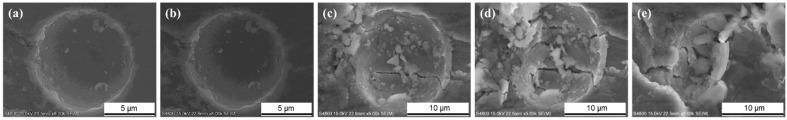
GHS fracture behavior of Al60G10 foam under compressive strain: (**a**) 1% strain. (**b**) 4% strain. (**c**) 8% strain. (**d**) 12% strain. (**e**) 15% strain.

**Figure 14 materials-16-02304-f014:**
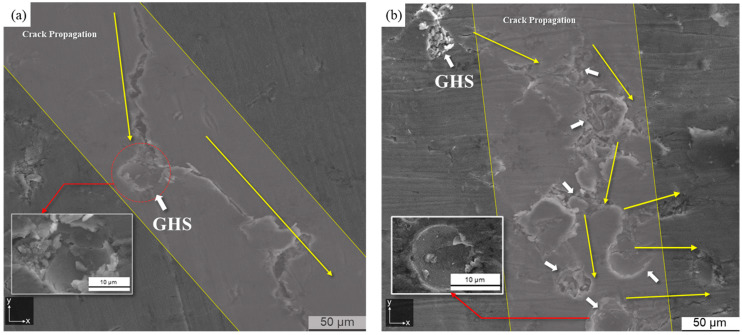
Crack propagation process; (**a**) Al60G5 and (**b**) Al60G10.

**Figure 15 materials-16-02304-f015:**
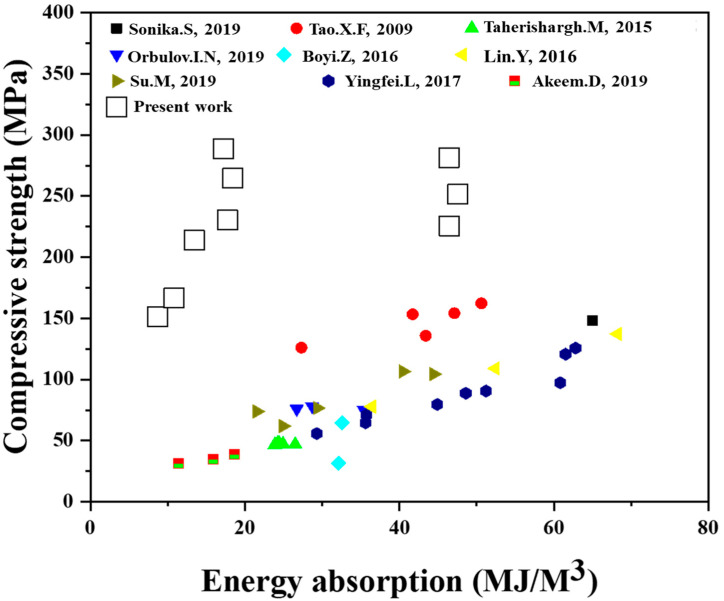
Comparison of results obtained from the present study with other literature data [6,41,44,47,48,49,50,51,52].

**Table 1 materials-16-02304-t001:** Al chips and GHS mixing ratio of AlG foam.

Specimen	Al Chip Volume Fraction (%)	GHS Volume Fraction (%)
Al50G5	50	5
Al50G10	50	10
Al50G15	50	15
Al60G5	60	5
Al60G10	60	10
Al60G15	60	15
Al70G5	70	5
Al70G10	70	10
Al70G15	70	15

**Table 2 materials-16-02304-t002:** Chemical composition of Al-8Zn-6Si-4Mg-2Cu alloys.

Elements	Zn	Si	Mg	Cu	Al
wt.%	7.94	5.99	4.01	1.98	Bal.

**Table 3 materials-16-02304-t003:** Chemical composition of GHS.

Elements	SiO_2_	Na_2_O	CaO	Al_2_O_3_	MgO	Etc.
wt.%	73.16	12.74	10.24	1.36	1.32	1.18

**Table 4 materials-16-02304-t004:** EDS results for raw GHS.

Elements	Si	O	Ca	Na	Mg	Etc.
wt.%	28.01	62.32	6.69	2.77	0.1	0.11

**Table 5 materials-16-02304-t005:** EDS results for GHS before sintering.

Elements	Si	O	Ca	Na	Mg	Al	Etc.
wt.%	28.05	61.20	6.37	3.96	0.14	0.11	0.17

**Table 6 materials-16-02304-t006:** EDS results for the GHS after sintering.

Elements	Al	Si	O	Mg
wt.%	33.89	1.19	52.92	12

**Table 7 materials-16-02304-t007:** Compressive properties of all specimens.

Specimen	Compressive Strength (MPa)	Plateau Strength (MPa)	Energy Absorption, W (MJ/M^3^)	Energy Absorption Efficiency $W_{e}\;(\%)$
Al chip	437.6 ± 1.2	489.8 ± 2.3	2.41 ± 1.1	0.49 ± 0.23
Al50G5	230.53 ± 4.4	292.71 ± 2.2	17.73 ± 3.7	6.0 ± 0.18
Al50G10	225.48 ± 4.7	377.82 ± 3.3	46.47 ± 4.1	12.29 ± 0.27
Al50G15	151.1 ± 6.1	173.21 ± 5.3	8.66 ± 2.4	5 ± 0.34
Al60G5	264.51 ± 8.1	315.13 ± 4.0	18.43 ± 4.2	5.8 ± 0.14
Al60G10	251.32 ± 6.7	387.24 ± 3.5	47.56 ± 3.5	12.28 ± 0.14
Al60G15	166.65 ± 5.3	210.31 ± 4.21	10.82 ± 3.1	5.1 ± 0.41
Al70G5	288.74 ± 5.2	338.72 ± 4.6	17.2 ± 5.4	5.07 ± 0.12
Al70G10	281.51 ± 6.4	411.14 ± 6.2	46.41 ± 2.2	11.2 ± 0.36
Al70G15	213.92 ± 3.8	251.13 ± 7.4	13.43 ± 4.6	5.3 ± 0.54

## Data Availability

Not applicable.

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
