# Peer review of "Synthesis and Characterization of Al Chip-Based Syntactic Foam Containing Glass Hollow Spheres Fabricated by a Semi-Solid Process"

_materials, 2023, doi:10.3390/ma16062304_

Round 1

Reviewer 1 Report

In this paper, aluminum (Al) chip matrix-based synthetic foams were fabricated through hot pressing at a semi-solid (SS) temperature. The interfacial reactions between the GHS and the Al chip alloy were investigated (by phase transformation) through thermodynamic simulations and microstructural characterization. The verification method considered is relatively complete, and the references are appropriate, including some useful research results, but there are still some contents that need to be improved:

1.    What is the substantive innovation of this work?

2.    How to control the pressure during specimen preparation, and what is the pressure corresponding to each heating period?

3.    For the mechanical property test, the size of the test piece is too small. How to consider the measurement deviation caused by size effect?

4.    Fig.11, the plastic strain of GHS10 Vol% is significantly higher than that of other cases. What is the reason?

Author Response

Response to Reviewer’s comments on the manuscript entitled “Synthesis and characterization of Al-chip based syntactic foam containing glass hollow spheres fabricated by a semi-solid process,” Manuscript ID: materials-2238532

The authors thank the editor and the reviewers for their careful reading and thoughtful comments on our paper, which helped significantly improve the manuscript. We have carefully considered the comments in preparing our new manuscript. The revised text is highlighted in yellow in the manuscript. The comments are in boldface in the following sections, while the authors' responses are in regular font.

Reviewer #1 :

  1. What is the substantive innovation of this work?

Response : This study is based on alloy fabrication that can improve both the energy absorption efficiency and compressive strength of Al-syntactic foam. In general, the Al-syntactic foam is known to have a high impact absorption efficiency but a low compressive strength (20-100 MPa) because of GHS. We fabricated a high-strength aluminum-based syntactic foam to improve the compressive strength of the Al syntactic foam. The prepared alloys have a compressive strength of 220-280 MPa and an energy absorption efficiency of 12%. These compressive properties are inferred to result from three strengthening effects throughout this study. The three strengthening effects are: Spheroidization of the microstructure inside the Al chip produced through the semi-solid process, strengthening of the matrix by the Mg2Si secondary strengthening phase, and improvement of the interface bonding between the matrix and GHS by the formation of spinel MgAl2O4 around the GHS. As a result, these strengthening effects produced an alloy with a higher compressive strength and energy absorption efficiency compared to common Al-syntactic foam. The synthesis and characterization of these alloys are the focus of this study.

  1. How to control the pressure during specimen preparation, and what is the pressure corresponding to each heating period?

Response : We controlled the pressure through hydraulics. We manually controlled the pressure by watching the pressure readings on the controller for each process. The process of applying pressure was pressurized to 120 MPa during a heating period in which the temperature was increased from 0 to 450 °C and held at 450 °C for 30 min. In the following process, the pressure was removed. We have clarified this in the revised manuscript (Figure. 1 (b)).

  1. For the mechanical property test, the size of the test piece is too small. How to consider the measurement deviation caused by size effect?

Response : We thank the reviewers for their valuable comments. We could examine the reproducibility and repeatability of the results by performing five compressive tests on each type of specimen five times. We have also indicated the standard deviation values for the compression test to account for measurement deviation. The additional discussion regarding this point can be found in Table 7 of the revised manuscript. The experimental method also indicates the number of compression test cycles (line 149 on page 5).

  1. Fig.11, the plastic strain of GHS10 Vol% is significantly higher than that of other cases. What is the reason?

Response : We believe that the higher plastic strain in the sample with 10 vol.% GHS is due to the appropriate GHS content. Samples with GHS 5 vol.% have a high compressive strength but a low plastic deformation rate because of stress localization in the Al matrix caused by the low ratio of GHS distribution. In contrast, the sample with 15 vol.% GHS has a high GHS content with low strength, which causes the collapse of the structure with a rapid drop in stress. The sample containing 10 vol.% GHS is characterized by uniform load transmission throughout the sample as a result of uniform stress distribution generated by the uniform distribution of GHS between the Al chip and the chip.

Reviewer 2 Report

Reviewer’s Comments:

The manuscript “Synthesis and characterization of Al chip based syntactic foam containing glass hollow spheres fabricated by semi-solid process” is very interesting work. In this study, aluminum (Al) chip matrix-based synthetic foams were fabricated through hot pressing at a semi-solid (SS) temperature. As a result, the glass hollow sphere (GHS) was uniformly distributed around the Al chip matrix, and a spherical microstructure, surrounded by the Mg2Si phase, formed in the interior. Mechanical characterization confirmed that the Al chip microstructure, consisting of the Mg2Si phase and GHS, exhibits high compressive strength and impact absorption capacity. Also, the Mg2Si phase and GHSs acted as load-bearing elements during compression and significant contributed to the compressive properties of the foam. To validate the presence of the interfacial reaction between the GHS and the matrix, an analysis was performed using an energy dispersive spectrometer.  However, the following issues should be carefully treated before publication.

1. In abstract, the author should add more scientific findings.

2. Keywords: the synthesized system is missing in the keywords. So, modify the keywords.

3. In the introduction part, the introduction part is not well organized and cited references should cite the recently published articles such as 10.1039/C9RA09349D and 10.1016/j.jallcom.2021.159013

4. Introduction part is not impressive and systematic. In the introduction part, the authors should elaborate the scientific issues in the syntactic foam research.

5. Results and discussion…, The author should provide reason about this statement “Fig. 4 shows the theoretical density of Al chips, Al chips mixed with GHS, and the relative density of Al foam by the volume fraction of GHS”.

6. The authors should explain regarding the recent literature why “he matrix of the Al chip alloy exhibited a hardness of 158.1 HV and did not appear to change with increasing GHS content”.

7. The author should explain the latest literature “The lower the volume fraction of Al powder, the shorter the distance between Al chips, and the GHS distributed over a relatively low Al powder area is determined to be agglomerated”.

8. The author should provide reason about this statement “The Al chip matrix was spheroidized by re-melting the Mg2Si phase during the reheating process”.

9. Comparison of the present results with other similar findings in the literature should be discussed in more detail. This is necessary in order to place this work together with other work in the field and to give more credibility to the present results.

10. The conclusion part is very week. Improve by adding the results of your studies.

Author Response

Response to Reviewer’s comments on the manuscript entitled “Synthesis and characterization of Al-chip based syntactic foam containing glass hollow spheres fabricated by a semi-solid process,” Manuscript ID: materials-2238532

The authors thank the editor and the reviewers for their careful reading and thoughtful comments on our paper, which helped significantly improve the manuscript. We have carefully considered the comments in preparing our new manuscript. The revised text is highlighted in yellow in the manuscript. The comments are in boldface in the following sections, while the authors' responses are in regular font.

Reviewer #2 :

  1. In abstract, the author should add more scientific findings.

Response : We have modified the abstract according to the valuable comments of the reviewers.

Before : In this study, aluminum (Al) chip matrix-based synthetic foams were fabricated through hot pressing at a semi-solid (SS) temperature. As a result, the glass hollow sphere (GHS) was uniformly distributed around the Al chip matrix, and a spherical microstructure, surrounded by the Mg2Si phase, formed in the interior. Mechanical characterization confirmed that the Al chip microstructure, consisting of the Mg2Si phase and GHS, exhibits high compressive strength and impact absorption capacity. Also, the Mg2Si phase and GHSs acted as load-bearing elements during compression and significant contributed to the compressive properties of the foam. To validate the presence of the interfacial reaction between the GHS and the matrix, an analysis was performed using an energy dispersive spectrometer. The results showed that MgAl2O4 was uniformly coated around GHS, which contributes, not only to the strength enhancement of the matrix, but also to the mechanical properties through the appropriate interfacial reactive coating.

After : In this study, aluminum (Al) chip matrix-based synthetic foams were fabricated by hot pressing at a semi-solid (SS) temperature. The densities of the produced Al chip-based syntactic foam ranged from 2.3 to 2.63 g/cm3, confirming that the density decreases with increasing glass hollow sphere (GHS) content. These values are approximately 16% lower than the density of Al chip alloys without GHS. The Al-chip syntactic foam microstructure fabricated by the semi-solid process consisted of GHS uniformly distributed around the Al-chip matrix and a spherical microstructure surrounded by the Mg2Si phase in the interior. The resulting spherical microstructure contributed significantly to the improvement of mechanical properties. Mechanical characterization confirmed that the Al chip syntactic foam exhibits compressive strength of approximately 225-288 MPa and an energy absorption capacity of 47 MJ/M3. These results indicate higher compressive properties than typical Al-syntactic foam. The Al-chip microstructure, consisting of the Mg2Si phase and GHS, acted as a load-bearing element during compression, contributing significantly to the compressive properties of the foam. An analysis was performed using an energy-dispersive spectrometer to validate the interfacial reaction between the GHS and the matrix. The results showed that MgAl2O4 was uniformly coated around GHS, which contributes not only to the strength enhancement of the matrix but also to the mechanical properties through the appropriate interfacial reactive coating.

  1. Keywords: the synthesized system is missing in the keywords. So, modify the keywords.

Response : We have edited the keywords according to the reviewers’ comments.

Before : High-strength aluminum alloy; semi-solid process; glass hollow sphere; microstructure; energy absorption efficiency

After : High-strength aluminum alloy; semi-solid; glass hollow sphere; uniaxial-pressure sintering; energy absorption efficiency

  1. In the introduction part, the introduction part is not well organized and cited references should cite the recently published articles such as 10.1039/C9RA09349D and 10.1016/j.jallcom.2021.159013

Response : Thank you for the reviewer's good remarks. We have changed the references cited in the Introduction to papers published in the last five years. This has been reflected in the revised manuscript.

Kannan, S.; Kishawy, H.A.; Pervaiz, S.; Thomas, K.; Karthikeyan, R.; Arunachalam, R. Machining of novel AA7075 foams containing thin-walled ceramic bubbles. Mater. Manuf. Process. 2020, 35, 1812–1821, doi:10.1080/10426914.2020.1802038.

Yingfei, L.; Qiang, Z.; Jing, C.; Haiyan, W.; Xiaowei, F.; Juan, W. Microstructural characterization and compression mechanical response of glass hollow spheres/Al syntactic foams with different Mg additions. Mater. Sci. Eng. A. 2019, 766, doi.org/10.1016/j.msea.2019.138338

Kartheek, S.M.; Vincent, S.; Suresh, K.R.N. Effect of single and hybrid hollow sphere reinforcement on the deformation mechanism of aluminum matrix syntactic foam at low strain rate. J. Alloys Compd. 2022, 901,  doi.org/10.1016/j.jallcom.2021.163573.

Mengxin, C.; Fengchun, J.; Chunhuan, G.; Yanchun, L.; Tianmiao, Y.; Ruonan, Q. Interface characterization and mechanical property of an aluminum matrix syntactic foam with multi-shelled hollow sphere structure. Ceramics international. 2022, 48, 18821–18833, doi.org/10.1016/j.ceramint.2022.03.159

Zhang, B.; Zhang, J.; Wang, L.; Jiang, Y.; Wang, W.; Wu, G. Bending behavior of cenosphere aluminum matrix syntactic foam-filled circular tube. Engineering Structures. 2021, 243, doi.org/10.1016/j.engstruct.2021.112650.

Sonti, K.S.M.; Vincent, S.; Narala, S.K.R. Quasi-static compressive response and energy absorption properties of aluminum matrix syntactic foams: Room temperature and elevated temperature conditions. Materials today communications. 2023, 35, doi.org/10.1016/j.mtcomm.2023.105580.

Katona, B.; Szlancsik, A.; Tabi, T.; Orbulov, I.N. Compressive characteristics and low frequency damping of aluminium matrix syntactic foams. Mater. Sci. Eng. A. 2019, 739, 140-148, doi.org/10.1016/j.msea.2018.10.014

Jung, J.; Kim, S.H.; Kang, J.H.; Park, J.; Kim, W.K.; Lim,C.Y.; Park,Y.H.; Compressive strength modeling and validation of cenosphere-reinforced aluminum-magnesium-matrix-based syntactic foams. Mater. Sci. Eng. A. 2022, 849, doi.org/10.1016/j.msea.2022.143452

  1. Introduction part is not impressive and systematic. In the introduction part, the authors should elaborate the scientific issues in the syntactic foam research.

Response : Thank you for the reviewer's good comments. The revised manuscript includes the introduction part and the scientific issues of the syntactic foam. We hope that the revised manuscript can be further taken into consideration for publication (line 67 on page 2).

  1. Results and discussion…, The author should provide reason about this statement “Fig. 4 shows the theoretical density of Al chips, Al chips mixed with GHS, and the relative density of Al foam by the volume fraction of GHS”.

Response : Thank you for your reviewer's comments. We have corrected the text in the revised manuscript to ensure there are no misunderstandings in the text (line 168 on page 5 and line 178 on page 6).

Before : The theoretical densities of AlG foam mixed with 100 vol.% Al chip alloy and 5, 10, and 15% GHS are 2.78, 2.66, 2.54, and 2.43 g/cm3, respectively. The relative density of AlG foam is reduced by about 6-16% compared to the theoretical density of the 100% Al chip alloy. These results indicate that the relative density of the AlG foam was reduced by the relatively light GHS mixture. The relative densities of AlG foams containing 5, 10 and 15 vol.% showed 98.8%, 98.7% and 96.2% relative to the theoretical density of the AlG foam. The relative density of the AlG foam is shown to be close to the theoretical density, and the partially reduced relative density of the 15 vol.% AlG foam can be explained as unintended microporous formation during the process.

After : The relative density of AlG foam is reduced by about 6-16% compared to the theoretical density of the 100% Al chip alloy. These results indicate that the relative density of the AlG foam was reduced by the relatively light GHS mixture. The relative densities of AlG foams containing 5, 10 and 15 vol.% showed 98.8%, 98.7% and 96.2% relative to the theoretical density of the AlG foam. The relative density of the AlG foam is shown to be close to the theoretical density, and the partially reduced relative density of the 15 vol.% AlG foam can be explained as unintended microporous formation during the process.

  1. The authors should explain regarding the recent literature why “he matrix of the Al chip alloy exhibited a hardness of 158.1 HV and did not appear to change with increasing GHS content”.

Response : We appreciate the reviewers' comments. We have revised the sentence for clarity (line 193 on page 6).

Before : Fig. 5(b) shows the matrix hardness of the GHS volume fraction. The matrix of the Al chip alloy exhibited a hardness of 158.1 HV and did not appear to change with increasing GHS content. The hardness values of Al50G5, Al60G5, and Al70G5 were 156.2, 156.1, and 155.8 HV, respectively. The hardness values of Al50G15, Al60G15, and Al70G15 were 156.3, 155.9, and 154.2 HV, respectively. The Al chip alloy and Al50, Al60, and Al70 hardness values show no change regardless of the volume fraction of GHS hence, GHS is not expected to have a significant effect on the mechanical properties of the foam.

After : Fig. 5(b) shows the evolutions of the hardness of the AlG foam with increasing volume fraction of GHS increased. Hardness decreased little with volume fraction of GHS. The matrix of the Al chip alloy shows a hardness of 158.1 HV and the matrix hardness of the AlG foam by GHS content is about 154.2-156.3 HV. The Al chip alloy matrix and the AlG foam matrix do not significantly differ in hardness. This is expected because of an appropriate interfacial reaction between the GHS and the Al matrix. The appropriate interfacial reaction between the GHS and matrix affects the mechanical properties [27,28]. Based on these results, it is presumed that the interfacial bonding between the GHS and the Al matrix of AlG foam is well done and the hardness is not reduced.

Liang-Jing, F.; Shueiwan, J. Reaction effect of Fly Ash with Al-3Mg melt on the microstructure and hardness of aluminum matrix composites. Materials and Design. 2015, 15, doi.org/10.1016/j.matdes.2015.10.070

Schultz, B.F.; Ferguson, J.B.; Rohatgi, P.K. M. Microstructure and hardness of Al2O3 nanoparticle reinforced Al-Mg composites fabricated by reactive wetting and stir mixing. Materials Science and Engineering A. 2011, 530, doi.org/10.1016/j.msea.2011.09.042.

  1. The author should explain the latest literature “The lower the volume fraction of Al powder, the shorter the distance between Al chips, and the GHS distributed over a relatively low Al powder area is determined to be agglomerated”.

Response : We thank the reviewers for their encouraging comments. We have revised the manuscript for clarity. (line 219 on page 7)

Before : The lower the volume fraction of Al powder, the shorter the distance between Al chips, and the GHS distributed over a relatively low Al powder area is determined to be agglomerated

After : . The lower the volume fraction of Al powder, the more GHS agglomerates within the area of Al powder for samples containing relatively high GHS volume fraction

Sudha, G.T.; Stalin, B.; Ravichandran, M.; Balasubramanian, M. Mechanical properties, characterization and wear behavior of powder metallurgy composites – A REVIEW. Materials today. 2020, 22, doi.org/10.1016/j.matpr.2020.03.389.

Canakci, A.; Varol, T. Microstructure and properties of AA7075/Al-SiC composites fabricated using powder metallurgy and hot pressing. Powder Technology. 2014, 268, doi.org/10.1016/j.powtec.2014.08.016 0032-5910/Crown

  1. The author should provide reason about this statement “The Al chip matrix was spheroidized by re-melting the Mg2Si phase during the reheating process”.

Response : We thank the reviewer for this excellent comment. We have identified a semantic transfer error in the sentence, "The Al chip matrix was spheroidized by re-melting the Mg2Si phase during the reheating process.” We have added the discussion in the revised paper (line 228 on page 7).

Before : The Al chip matrix was spheroidized by re-melting the Mg2Si phase during the reheating process.

After :  The microstructure of the Al chip matrix was spheroidized according to the SS process. Previous studies have confirmed that the Al5Cu2Mg8Si6 phase becomes the liquid phase via DSC at about 450 - 470 °C. Small amounts of liquid are essential in the spheroidization of the Al chip matrix. Under 120 MPa pressure, a small volume of liquid is squeezed onto the surface of the dendrite. The Al matrix is then spheroidized by rheology and the Ostwald ripening phenomenon after a longer holding time at 560 °C.

Son, Y.G.; Jung, S.S.; Park, Y.H.; Lee, Y.C. Effect of semi-solid processing on the microstructure and mechanical properties of aluminum alloy chips with eutectic Mg2Si intermetallics. Metals (Basel). 2021, 11, doi:10.3390/met11091414.

Yong, H.; Sheng-qi, F.; Long-zhi, Z.; Da-hao, W.; Fei, L. Microstructure evolution of semi-solid Mg2Si/A356 composites during remelting process. Research & Development. 2020, 17, doi.org/10.1007/s41230-020-9158-7.

  1. Comparison of the present results with other similar findings in the literature should be discussed in more detail. This is necessary in order to place this work together with other work in the field and to give more credibility to the present results.

Response : In order to response to this point, we have added an additional figure (Fig.15 in the revised manuscript) which shows the energy absorption versus compressive strength of Al syntactic foam in this study and other researches.

Fig. 15 shows the energy absorption versus compressive strength of the Al-syntactic foam in this and other studies. The AlG foam with 10 vol.% synthesized in this study has the highest compressive strength and the third highest energy absorption rate among those reported in the literature. However, AlG foams containing 5 and 10 vol.% have high compressive strengths but normal energy absorptions. These results mean that it is possible to synthesize materials with high compressive strengths and energy absorptions of the Al foam. It is also expected that a uniaxial pressure sintering system can achieve the production of functional materials whose stability is ensured.

  1. The conclusion part is very week. Improve by adding the results of your studies.

Response : We thank the reviewers for the valuable comments. We have edited the conclusion. (line 392 of page 14)

Before : In this study, an Al-chip (spheroidized) matrix-based syntactic foam mixed with GHS was successfully synthesized using a uniaxial pressurized sintering process. The average density of the synthesized foam was reduced by up to 6-16% compared with that of the Al chip alloy in the present study. The interfacial reactions between the GHS and the Al chip alloy were investigated (by phase transformation) through thermodynamic simulations and microstructural characterization. Al and Mg formed in the GHS shell by the phase transformation to MgAl2O4 through the reaction with SiO2 in the GHS with a composite chemical composition. The spinel MgAl2O4 formed during the interfacial reaction between the GHS and the matrix has the potential to strengthen the interfacial bond. Furthermore, while MgAl2O4 formed, the Si solute was released, by the reduction of GHS, and reacted with Mg in the matrix to form a Mg2Si phase around the GHS shell. The precipitation of Mg2Si improved the mechanical properties of the foam. Al chip matrix-based foams mixed with GHS improve the compressive properties and energy absorption simultaneously because of three reinforcing effects: spheroidized Al chip matrix, spinel MgAl2O4 by interfacial reaction, and Mg2Si secondary phase.

After : 1. The semi-solid process using uniaxial-pressure sintering has successfully produced Al chip-based syntactic foam with a volume fraction of up to 15% of GHS. The microstructure of spheroidized Al chips produced by the semi-solid process is characterized by higher strength than that of ordinary Al. For uniaxial pressure sintering, the temperature is raised from 0 to 450 °C and 120 MPa is applied during sintering at 450 °C for 30 min. Then, after removing the pressure from the process, an Al chip-based syntactic foam could be produced with a sintering temperature of 560 °C and a sintering time of 30 min.

  1. The average density of the produced foam was up to 16% lower than that of the Al chip alloy without GHS. It also has a relative density that is close to a maximum of 98% relative to the theoretical density of the synthesized foam. GHS were distributed between the Al chip and the chip in the microstructure, and uniform spheroidization of the Al chip matrix was observed. The spheroidization of the matrix appears to be caused by a small amount of liquid being squeezed onto the dendrite surface, followed by an Ostwald ripening mechanism.
  2. The interfacial reactions between the GHS and the Al chip alloy were investigated (by phase transformation) by thermodynamic simulations and microstructural characterization. Aluminum and magnesium were formed in the GHS shell by the phase transformation to MgAl2O4 through the reaction with SiO2 in the GHS with a composite chemical composition. The spinel MgAl2O4 formed during the interfacial reaction between the GHS and the matrix can potentially to strengthen the interfacial bond. Furthermore, while MgAl2O4 was being formed, the Si solute was released, by the reduction of GHS and it reacted with Mg in the matrix to form a Mg2Si phase around the GHS shell. The precipitation of Mg2Si improved the mechanical properties of the foam.
  3. Foam containing 10 vol.% volume fraction of GHS in compressive properties exhibits compressive strengths of up to 288 MPa and energy absorption of 47 MJ/M3. These results show improved compressive properties and energy absorption compared to conventional Al syntactic foams due to the three reinforcing effects of the spheroidized Al chip matrix, spinel MgAl2O4 by interfacial reaction, and Mg2Si as the secondary phase. On the other hand, foam containing a 15 vol.% volume fraction of GHS shows low compressive properties because the agglomerates of the GHS do not result in uniform load transfer.

Reviewer 3 Report

Recommendation: Major revisions needed as noted.

Comments:

In this manuscript, glass hollow sphere (GHS) was used to act as dopant in aluminum (Al) chip matrix-based syntactic foam to increase the mechanical properties of Al chip. The author tests the compressive strength and energy absorption capacity and validate the interfacial reaction between GHS and metal matrix. Overall, the idea is interesting. However, there is no significant innovations on the material, the fabrication process nor the performance. The previous study published on <Metals>, 2021. Besides, this paper lacks several aspects of analysis and results. Here is the list of suggestions for the author needs to consider:

1.       The author listed other researcher’s work in the Introduction section. But they lack the comparison between the idea of this study and other researcher’s work. From the statement in Page 2, I didn’t see enough innovations in this work compared to others. The author should emphasize the key innovative ideas in this section.

2.       In Fabrication Section, the author should mention how they originally machined Al chips and the size distribution. This is a critical data for the following discussion.

3.       In fabrication process, since the material is temperature sensitive, it is better to provide the heater temp distribution map over the heater plate.

4.       Figure 1 (b) and (c) can be combined in one figure.

5.       In Results and Discussion Section. The author states their observations by figures. For example: Fig. 3 shows… Fig. 4 shows…

For a scientific publication, the aim of the figures is to support your hypothesis or conclusion. In this situation, the statement should point out your goal or hypothesis. Then, in order to validate your thoughts, you did the experiments and attach the results. And then explain how the results support your thoughts.

6.       References are not enough to cover the state-of-art research in this topic.

Thanks.

Author Response

Response to Reviewer’s comments on the manuscript entitled “Synthesis and characterization of Al-chip based syntactic foam containing glass hollow spheres fabricated by a semi-solid process,” Manuscript ID: materials-2238532

The authors thank the editor and the reviewers for their careful reading and thoughtful comments on our paper, which helped significantly improve the manuscript. We have carefully considered the comments in preparing our new manuscript. The revised text is highlighted in yellow in the manuscript. The comments are in boldface in the following sections, while the authors' responses are in regular font.

Reviewer #3 :

  1. The author listed other researcher’s work in the Introduction section. But they lack the comparison between the idea of this study and other researcher’s work. From the statement in Page 2, I didn’t see enough innovations in this work compared to others. The author should emphasize the key innovative ideas in this section.

Response : In order to response to this point, we have added an additional figure (Fig.15 in the revised manuscript) which shows the energy absorption versus compressive strength of Al syntactic foam in this study and other researches.

Fig. 15 shows the energy absorption versus compressive strength of Al syntactic foam in this study and other researches. The AlG foam with 10 vol.% synthesized in this study has the highest compressive strength and the third highest energy absorption rate among those reported in the literature to date. However, AlG foams containing 5 and 10 vol.% have high compressive strength but normal energy absorption. These results mean that it is possible to synthesize materials with high compressive strength and energy absorption of Al foam. It is also expected that a uniaxial pressure sintering system can achieve the production of functional materials whose stability is ensured.

  1. In Fabrication Section, the author should mention how they originally machined Al chips and the size distribution. This is a critical data for the following discussion.

Response : The aluminum alloy was processed through a metal crusher. We sieved with a 500 µm-sized sieve to obtain uniformly sized aluminum chips. This information is currently added to line 101 on page 3 of the revised manuscript.

Before : The AlG foam matrix was designed by melt casting an Al-8Zn-6Si-4Mg-2Cu alloy, and the fabricated Al alloy was machined into Al chips with a width and length of approximately 1 mm or less (Fig. 1). AlG foam were synthesized using a uniaxial-pressure sintering process.

After: The AlG foam matrix was designed by melt casting an Al-8Zn-6Si-4Mg-2Cu alloy, and the fabricated Al alloy was machined into Al chips with a width and length of approximately 1 mm or less (Fig. 1). Al chips were processed in a metal crusher. It was sieved with a 500 µm sieve to obtain uniformly sized al chips. AlG foam was synthesized using a uniaxial-pressure sintering process.

  1. In fabrication process, since the material is temperature sensitive, it is better to provide the heater temp distribution map over the heater plate.

Response : We think it would be a good practice to provide measured ambient temperatures per the reviewer's opinion. In this experiment, measuring the actual ambient temperature was considered a limitation in controlling the optimal process temperature. Therefore, we measured the internal temperature at the mold closest to the sample for a more reliable temperature measurement. As a result, we confirmed that our designed temperatures matched the mold temperatures.

  1. Figure 1 (b) and (c) can be combined in one figure.

Response : We thank the reviewers for the valuable comments. We have edited the figure as follows.

This information is added to line 128 on page 4 of the revised manuscript.

  1. In Results and Discussion Section. The author states their observations by figures. For example: Fig. 3 shows… Fig. 4 shows…

For a scientific publication, the aim of the figures is to support your hypothesis or conclusion. In this situation, the statement should point out your goal or hypothesis. Then, in order to validate your thoughts, you did the experiments and attach the results. And then explain how the results support your thoughts.

Response : We have modified the manuscript according to the valuable comments of the reviewers (line 168 on page 5, line 193 on page 6).

  1. References are not enough to cover the state-of-art research in this topic.

Response : We agree with the reviewer that the original manuscript lacks references to address the state-of-the-art research on this topic. We tried our best to improve our manuscript in the limited time of the revision process. The revised manuscript has more information and detailed analysis. We hope that the revised manuscript can be further considered for publication.

Round 2

Reviewer 3 Report

After major revision, this manuscript is ready to be published in the current version.